Journal of
open psychology data

# Data From Early Childhood Educators' Work and Stress Study

DATA PAPER

**RANDI A. BATES** iD

**JACLYN M. DYNIA** iD

**BAILEY E. MARTIN** iD

*Author affiliations can be found in the back matter of this article

]u[ ubiquity press

## ABSTRACT

This paper describes a longitudinal dataset of perceived job stress, perceived general stress, financial stress, demographics, and educational center characteristics of center-based early childhood educators in the Midwest across the academic year 2021–2022. At four time points, a convenience sample of 67 educators completed electronic surveys. At the first two time points, a subset provided hair cortisol samples to estimate physiological chronic stress. The publicly available, de-identified data can provide nuanced research and teaching opportunities into educators' stressors during a dynamic period of the COVID-19 pandemic.

**CORRESPONDING AUTHOR:**
**Randi A. Bates**

University of Cincinnati, United States

batesri@ucmail.uc.edu

**KEYWORDS:**
Early childhood education; longitudinal; stress; work; job

**TO CITE THIS ARTICLE:**

# (1) BACKGROUND

Alongside parents and other caregivers, early childhood educators (for conciseness, hereafter called "educators") are essential in rearing over 33% of young children (0-5 years old) in the United States (U.S.; U.S. Department of Education, 2021). Educators enable children's families to return to the labor force, and many educators may spend more awake/interactional time with children than children's parents or legal guardians (National Center for Education Statistics, 2023; U.S. Department of Education, 2021). Additionally, high-quality early childhood education programs are an evidence-based intervention to buffer the adverse effects of poverty on children's development (Shonkoff & Fisher, 2013).

However, emerging evidence suggests that many educators may be experiencing inequitably high levels of stress compared to other adults in the United States (Linnan et al., 2017; Otten et al., 2019; Whitaker et al., 2015). Sources of educators' stress, otherwise known as stressors, may be related to earning less than 98% of other U.S. occupations (McLean et al., 2021) while managing increasing levels of intergenerational family trauma, conflicts with families, and a lack of occupational equipment for adult-sized bodies (Bates & Dynia, 2023; Friedman-Krauss et al., 2014). In turn, this stress may have reverberating costs on all of society, leading to morbidity and mortality in educators (Ford et al., 2004; McLaughlin, 2011), unsustainable turnover in early childhood educational workplaces (Grant et al., 2019), and poor socioemotional development in children (Schonert-Reichl, 2017) that may increase children's risk of expulsion and lifelong adverse well-being (Hansen & Broekhuizen, 2021; Hatfield et al., 2016; Jennings & Greenberg, 2009; Lisonbee et al., 2008; Mashburn et al., 2008; Miller et al., 2023; Moffitt et al., 2011; Shin, 2010). However, the relationship between educators' stressors and stress is poorly understood.

## 1.1 THEORETICAL BACKGROUND OF EDUCATORS' STRESS

Broadly, stress is a psychophysiological reaction to a perceived stressor. Conceptualizing educators' stress is inspired by two complementary but distinct theories: the job demands and resources (JD-R) model (Bakker & Demerouti, 2007), adapted for early childhood educators (Kwon et al., 2021), and the allostatic load theory of stress (McEwen & Wingfield, 2010). The educator-adapted JD-R model generally posits that if educators' job demands exceed job resources, then stress may result (Kwon et al., 2021). The allostatic load theory of stress suggests that while stress is necessary for survival, repeated stressors may result in "wear and tear" on the body (McEwen & Wingfield, 2010). Further, psychological and physiological stress responses may differ from each other to keep the body in homeostasis (McEwen

& Wingfield, 2010). Research has also shown that the timing of stressors and occupational burnout may influence workers' physiological stress responses (Bärtl et al., 2023). As such, it is necessary to comprehensively examine potential sources and buffers of educators' stress, including their psychological and physiological stress responses, over time.

## 1.2 OBJECTIVE AND THE PRESENT DATASET

To help define the relationship between work-related factors and educators' stress, as well as identify modifiable targets for intervention, the objective of this longitudinal study and dataset was to provide a rich resource to examine work-related factors and stress in early childhood educators over time. Due to the distinct work experiences between center- and home-based educators, this dataset focused on center-based early childhood educators. The dataset allows researchers to examine the stress and stressors of a sample of center-based educators over four time points across an academic year. The sampling period overlapped with the unexpected Omicron surge of the COVID-19 pandemic, which resulted in unexpected and immediate classroom closures in several educational centers. As such, the temporality of the dataset also allows researchers to build causal evidence towards distinct changes in workplace-related stressors and resulting educators' stress.

## 1.3 MAJOR USES OF THE DATA

To date, the data have been used to describe educators' personal and physiological stress at the beginning of the academic year (Bates & Dynia, 2023, 2024a). Gaps remain in examining work-related stressors and stress across the academic year.

# (2) METHODS

## 2.1 STUDY DESIGN

This was a longitudinal, observational study on job characteristics and stress of center-based early childhood educators. Across four time points of the academic year 2021–2022, educators completed electronic surveys on perceived job stress, perceived general stress, financial stress, their demographics, and educational center characteristics. At the first two time points, a subset provided hair samples for cortisol analysis as an estimate of chronic, physiological stress.

## 2.2 TIME OF DATA COLLECTION

Data for time point one were collected from August to November 2021. Time point two data were collected from December 2021 to April 2022. Time point three data were obtained from March to May 2022. The last time point of data, time point four, were collected from June to July 2022.

## 2.3 LOCATION OF DATA COLLECTION

Educators completed electronic surveys at their convenience via Qualtrics and REDCap (Harris et al., 2019; Harris et al., 2009). Hair samples were also collected at educators' convenience: at their early childhood educational center or home located near central and southwest Ohio in the United States of America.

## 2.4 SAMPLING, SAMPLE AND DATA COLLECTION

Center-based early childhood educators were recruited using a targeted purposive sampling strategy of those working through community partners with whom we had existing relationships. We distributed flyers and emails about the study through listservs, administrators, and early childhood educators. Eligibility criteria included (a) at least 18 years old and (b) employed primarily as a center-based educator of children 0–5 years old.

After eligibility was confirmed, we collected electronic consent. Educators were compensated with a $10 grocery store gift card for completing each survey and a $15 grocery store gift card for providing each hair sample. Educators could earn up to $70 in compensation for participating.

Initially, 71 educators consented to participate. Of these, four educators never completed any study measures, and one immediately withdrew after opening the first survey. As such, the initial sample size at time point one consisted of 67 center-based early childhood educators from 13 educational centers. The participation of the educators by time point is in Table 1. Educators' demographics at enrollment are in Table 2.

In Table 1, readers can see that while formal withdrawals were few, there were more non-responders who did not complete surveys across TP1-3. While it is not directly known why some participants were non-responders, TP2 and 3 overlapped with the highly disruptive COVID-19 Omicron variant, which may have affected participation.

## 2.5 MATERIALS/SURVEY INSTRUMENTS

The materials for this study include survey data and data on hair cortisol analysis. Details of non-copyrighted study measures, including around 550 items, the respective scoring (including blinded or deleted variables to protect participant privacy), and when measures were assessed, can be found in the published codebook (Bates & Dynia, 2024b). Overall, we gathered data from six major constructs to understand early childhood educators' stress: (1) work characteristics; (2) personal demographics; (3) socioeconomic stress or stressors; (4) perceived personal stress or stressors, (5) perceived job stress or stressors, and (6) physiological stress. Psychometrics for standardized instruments are available from their respectively cited, published manuscripts.

### Work Characteristics

Work characteristics were measured by several researcher-developed questions surrounding several concepts. These concepts included educators' credentials (e.g., major for highest degree), the educators' work center characteristics (e.g., center accreditation), their personal work characteristics (e.g., hourly wages), and job turnover (e.g., whether they changed their job in the past three months).

### Personal Demographics

Educators' personal demographics were measured by commonly used questions assessing educators' annual household income and common sociodemographics such as highest level of education, year born, gender, race/ethnicity, and language spoken.

### Sociodemographic Stress or Stressors

Sociodemographic stress or stressors were measured with questions surrounding housing concerns (items from the Preschool Promise Study; Purtell et al., 2021), institutional resources and economic hardship (Yoshikawa et al., 2008), economic hardship (items from the Preschool Promise Study; Purtell et al., 2021), and food insecurity (Blumberg, 1999). More details about the standardized Yoshikawa Institutional resources, economic hardship, and food insecurity scales follow.

**Institutional resources and economic hardship.** Institutional resources and economic hardship were assessed with a modified version of the Yoshikawa et al. (2008) instrument. For institutional resources, three items were assessed: whether the educator or someone in their household had a checking account, savings account, or driver's license. Items were scored 1 = *yes* or 0 = *no*. The possible range was from 0–3, with higher sum scores indicating more institutional resources. For

| | TP1 | | TP2 | | TP3 | TP4 |
|---|---|---|---|---|---|---|
| | **SURVEY** | **HAIR** | **SURVEY** | **HAIR** | **SURVEY** | **SURVEY** |
| **Participants** | 67 | 50 | 53 | 41 | 45 | 42 |
| **Formal withdrawals** | 1 | | 2 | | 0 | 0 |

**Table 1** Formal Participation of the Sample in Surveys and Hair Samples by Time Point (TP).

*Note.* Hair was only collected at time point 1 and 2.

| | % | *M* | *SD* | RANGE |
|---|---|---|---|---|
| **Annual household income** | | | | |
| $10,000 or less | 7.94 | | | |
| $10,001 to 20,000 | 4.76 | | | |
| $20,001 to 30,000 | 25.4 | | | |
| $30,001 to 40,000 | 22.22 | | | |
| $40,001 to 50,000 | 12.7 | | | |
| $50,001 to 60,000 | 6.35 | | | |
| $60,000 or more | 20.63 | | | |
| **Education** | | | | |
| High school diploma or equivalent, plus technical training or certificate | 13.85 | | | |
| Some college but no degree | 21.54 | | | |
| Associates degree | 13.85 | | | |
| Bachelor's degree | 36.92 | | | |
| Master's degree | 7.69 | | | |
| Education specialist or professional diploma based on at least one year of course work beyond a Master's degree | 3.08 | | | |
| Doctoral degree | 1.54 | | | |
| Other | 1.54 | | | |
| **Gender female** | 95.2 | | | |
| **Language spoken fluently other than English** | 4.5 | | | |
| **Race** | | | | |
| Black/African American | 29.85 | | | |
| American Indian or Alaska Native | 0 | | | |
| White/Caucasian | 68.66 | | | |
| Native Hawaiian or Other Pacific Islander | 1.49 | | | |
| Asian | 1.49 | | | |
| Other | 0 | | | |
| **Ethnicity Hispanic or Latino** | 3.13 | | | |
| **Year born** | | 1981.57 | 13.48 | 1952–2002 |

**Table 2** Educators' Demographics at Enrollment (n = 67).

*Note.* Percentages compiled from respondents. Respondents could select more than one race.

economic hardship, educators were asked four items: if there was a time in the past year that they and their family did not have telephone service, did not pay the full rent or mortgage, were evicted for not paying their rent or mortgage, or lost service from utility companies due to non-payments. Items were scored 1 = *yes* or 0 = *no*. Total possible scores were from 0–4, with higher scores indicating more economic hardship.

**Food insecurity.** Educators' food insecurity in the past 6 months was assessed with a modified 6-item self-report short-form food insecurity questionnaire from the US Department of Agriculture (Blumberg, 1999; United States Department of Agriculture [USDA], 2012). Information about modifications is found in the

codebook. Items were scored 1 = *yes* or 0 = *no*, for a possible sum score from 0–6. Higher scores indicate more food insecurity, with raw scores of 0–1 indicating food security and scores of 2–6 indicating food insecurity (United States Department of Agriculture [USDA], 2012).

## Perceived Personal Stress or Stressors

Perceived personal stress or stressors were measured by several concepts. These included personal stress (Personal Stress Scale; Cohen et al., 1983), personal self-efficacy (generalized self-efficacy scale; Schwarzer & Jerusalem, 1995), depression (Center for Epidemiologic Studies–Depression Scale; Radloff, 1977), anxiety (generalized anxiety disorder-7 scale; Spitzer et al., 2006),

and childhood stressors (adverse childhood experiences; Center for Youth Wellness, 2015).

**Personal Stress.** Personal stress was measured with the Perceived Stress Scale (Cohen et al., 1983), a 10-item self-report questionnaire on stress from the past month. Educators rated items on a 5-point frequency scale: 0 = *never*, 1 = *almost never*, 2 = *sometimes*, 3 = *fairly often*, and 4 = *very often*. Possible sum scores are from 0–40, with higher scores suggesting higher perceived general stress.

**Personal self-efficacy.** General self-efficacy was measured with the Generalized Self-efficacy Scale (Schwarzer & Jerusalem, 1995). The instrument includes 10 self-report items rated on a 4-point Likert scale, scored 0 = *not at all true*, 1 = *hardly true*, 2 = *moderately true*, and 3 = *exactly true*. Possible sum scores are from 0–30, with higher scores suggesting higher self-efficacy.

**Depression.** Depression was measured with the self-report using the Center for Epidemiologic Studies Depression Scale (CES-D) from Radloff (1977). Educators rated 20 items on a 4-point frequency scale of depressive related feelings in the past week. Items were scored 0 = *rarely or none of the day (less than 1 day)*, 1 = *some or a little of the time (1–2 days)*, 2 = *occasionally or a moderate amount of time (3–4 days)*, and 3 = *most or all of the time (5–7 days)*. Possible scores are from 0–60, with higher scores indicative of higher depressive symptoms.

**Anxiety.** Anxiety was measured with the General Anxiety Disorder-7 (GAD-7) scale (Spitzer et al., 2006). Educators rated 7 self-report items on a 4-point frequency scale on anxiety-related problems in the past 2 weeks. Items were scored 0 = *not at all*, 1 = *several days*, 2 = *more than half the days*, 3 = *nearly every day*. Possible sum scores are from 0–21, with higher values indicating higher anxiety levels.

**Childhood stressors.** Childhood stressors were assessed with a de-identified count of the Adverse Childhood Experiences Questionnaire (Felitti et al., 1998) adapted by the Center for Youth Wellness (2015). Educators were prompted to count how many of the adverse childhood experiences they experienced before reaching their 18th birthday. The first group of adverse childhood experiences was aligned with previous studies (e.g., Felitti et al., 1998; Wade et al., 2017) and datasets such as the 2021 Centers for Disease Control Behavioral Risk Factor Surveillance System (BRFSS). The second group of adverse childhood experiences was designed by the Center for Youth Wellness (2015). Details of these items are found in the codebook. Possible scores for the first group are from 0–10 and for the second group from 0–9, with higher scores indicating more adverse childhood experiences.

## Perceived Job Stress or Stressors

Perceived job stress or stressors were also measured by several concepts. These included job satisfaction, job coping, teacher beliefs or self-efficacy, teacher stress, teacher burnout, and perceived job satisfaction.

**Job Satisfaction.** Teachers' job satisfaction was assessed with (a) a modified version of the Work Attitudes Questionnaire (WAQ; Jorde-Bloom, 1988) that was used in the Kentucky Professional Development Framework Study (Rous & Grisham Brown, 2010), (b) three items asking educators to rank or state open-ended perspectives of job satisfaction or frustration from the Kentucky Professional Development Framework Study (Rous & Grisham Brown, 2010), and (c) three items from the Cincinnati Preschool Promise study (Purtell et al., 2021). Details of the non-standardized questions are found in the codebook, and details for the standardized WAQ follow.

The WAQ contains six subscales on (1) co-worker relations, (2) supervisor relations, (3) the nature of the work itself, (4) working conditions, (5) pay and promotional opportunities, and (6) characteristics of their ideal job. The first five subscales have 10 items each and are rated on a 5-point Likert scale from 0 = *strongly disagree* to 4 = *strongly agree*. Some items are reverse-scored. Possible subscale scores are from 0 to 40, with higher scores indicating higher job satisfaction. The sixth subscale on ideals has 5 items rated on a 5-point scale from 0 = *not like my ideal at all* to 4 = *exactly like my ideal*. The ideal subscale items are summed with possible scores from 0 to 20; higher scores indicate higher congruency with one's ideal job.

**Job coping.** Teachers' job coping was assessed with one item: "How well are you coping with the stress of your job right now?" (Herman et al., 2017). The item is scored on an 11-point scale from 0 = *not well* to 10 = *very well*.

**Teacher Beliefs or Self-Efficacy.** Teacher beliefs or self-efficacy were measured with the Teachers' Sense of Efficacy Scale-Short Form (Tschannen-Moran & Hoy, 2001). Educators rate 12 items across three subscales on their perceived efficacy in student engagement (4 items), instructional strategies (4 items), and classroom management (4 items). Items are rated on a 9-point agreement scale from 0 = *not at all* to 8 = *a great deal*. Items are averaged across subscales with possible subscale scores from 0 to 36. Higher scores indicate higher self-efficacy.

**Teacher stress.** Educators' stress was measured with the Teacher Stress Inventory (Fimian, 1984; Fimian & Fastenau, 1990). Educators completed 49 items across 10 subscales on aspects of teacher-related stress. Each subscale includes a varying amount of items and address (1) time management (8 items), (2) work-related stressors (6 items), (3) professional distress (5 items), (4) discipline and motivation (6 items), (5) professional investment (4 items), (6) emotional manifestations (5 items), (7) fatigue manifestations (5 items), (8) cardiovascular manifestations (3 items),

(9), gastronomical manifestations (3 items), and (10) behavioral manifestations (4 items). Educators rated each item on a 1–5 rating scale from 1 = *no strength; not noticeable* to 5 = *major strength; extremely noticeable*. Subscale items are averaged, and the total score is the sum of the subscale scores divided by 10. Higher scores suggest higher teacher stress.

**Teacher Burnout.** Educators' burnout was measured with the copyrighted Maslach Burnout Inventory – Educators Survey (MBI-ES; Maslach & Jackson, 1981). Educators assessed burnout-related feelings about their job with 22 items across three subscales: emotional exhaustion, personal accomplishment, and depersonalization. Educators rated the items using a 7-point frequency scale from 0 = *never* to 6 = *everyday.* Items are summed with possible scores ranging from 0 to 132, with higher scores indicating higher teacher burnout.

### Physiological Stress

Physiological stress was measured with the concepts of longitudinal physiological stress (hair cortisol) and chronic physiological stress confounders (Bates et al., 2020).

**Longitudinal Physiological Stress.** Chronic physiological stress was estimated with hair cortisol concentration in educators who chose to provide hair samples. Briefly, trained research nurses snipped a shoelace-tip-diameter hair sample adjacent to the skin of the posterior vertex portion of the scalp. These samples were stored at room temperature prior to laboratory processing. In the lab, the samples were cut to a 3cm length, most proximal to the scalp, to represent the last 3 months of hair growth at the time of sampling (Loussouarn et al., 2016). Following established methods (Meyer et al., 2014), the samples were ground to a powder then assayed in duplicate for cortisol concentration with Salimetrics® (n.d.) immunoassay kit. The average intra-assay coefficient of variation (CV) was 6.7%, and the inter-assay CV was 6.0% (<10% is considered acceptable). To normalize distributions, hair cortisol values are typically transformed from pg/mg to log10 (pg/mg). Higher values suggest higher cortisol levels.

**Chronic Physiological Stress Confounders.** To help understand the impact of major confounders on hair cortisol, educators also completed a survey on hair care and steroid use inspired by Bates et al. (2020).

### 2.6 QUALITY CONTROL

Several factors were used as quality control of both the survey and hair cortisol data during the study. For the survey data, experienced data managers created the electronic surveys in well-established platforms (Qualtrics and REDCap), which were pilot-tested several times before collecting data from the sample. In most circumstances, survey data were collected using multiple-choice options. For free text responses, quality checks from the software were not included.

During data collection, the research team checked for potential issues. One issue that arose during TP1 was regarding access to university licensed software. As a result, the team quickly switched from Qualtrics to collecting data in REDCap, for which the university had more complete access through the university Center for Clinical & Translational Science and Training. Consequently, data from TP1 were collected in Qualtrics and REDCap. That is, at TP1, all participants completed a short survey in Qualtrics and another in REDCap. Data from TP2–4 were only collected in REDCap.

For data cleaning, experienced data cleaners were hired to clean the data, which the team thoroughly reviewed and verified prior to analysis and publication. The paid data cleaners were independent from the investigator and data collection team. During cleaning, Qualtrics and REDCap data collected in TP1 were merged by adding cases to a common Excel spreadsheet. All data were transferred to an Excel spreadsheet and SPSS for data cleaning by each TP. Identifiable and impossible free text data values in the sample (e.g., hourly wages of $10,000), along with total scores with more than 10% of responses missing, were deleted from cells and marked as blinded or missing data.

For the hair cortisol data, trained registered nurses collected the hair samples and the principal investigator (PI) monitored random hair samples for appropriate sampling technique and size. The hair samples were analyzed by a professional lab that was well-experienced in hair cortisol analysis using established standards, as described earlier. During laboratory analysis, trained technicians analyzed the samples, resulting in low coefficients of variation (ideal coefficients of variation are <10%).

### 2.7 DATA ANONYMIZATION AND ETHICAL ISSUES

Ethical approval for the study was obtained from the University of Cincinnati Institutional Review Board (#2021–0567). Educators provided electronic informed consent on REDCap before completing any study activities.

To protect educators' confidentiality, several steps were taken. Educators were assigned a numerical ID, which they entered when completing surveys. This ID number was kept separate from educators' names and other identifiable information (e.g., contact information). We removed any identifiable information from variables containing free text responses. We also removed some demographics from the final dataset with few categories of educators (e.g., gender, as indicated in the codebook). All removed variables or blinded information to protect

participants' accidental identification are marked in the data dictionary section in the codebook. Finally, we followed several steps in cleaning data by independent researchers, which the PI checked multiple times before publicly publishing the dataset.

## 2.8 EXISTING USE OF DATA

To date, the data have been used for three published, peer-reviewed research manuscripts. These manuscripts are:

1. Bates, R. A. & Dynia, J. M. (2024a). Changes in stress following wage increases for early childhood educators. *Early Childhood Education Journal* doi: 10.1007/s10643-024-01666-0
2. Bates, R. A., Dynia, J. M. (2023). Psychological and physiological stress and stressors in early childhood educators: A pilot study. *Psychology in the Schools.* https://doi.org/10.1002/pits.23118
3. Bates, R. A., Almallah, W., Martin, B. E., Ananzeh, T. I., Collen, C., & Dynia, J. M. (2025). Feasibility of collecting hair for cortisol analysis in early childhood educators. *Nursing Research.* doi: 10.1097/NNR.0000000000000815

## (3) DATASET DESCRIPTION AND ACCESS

Files being shared are the data dictionary with codebook and four cleaned and de-identified data files: Time Point 1, Time Point 2, Time Point 3, Time Point 4, and Hair Cortisol for Time Point 1 and 2.

### 3.1 REPOSITORY LOCATION

The data and codebook are published on the LDbase repository here: https://doi.org/10.33009/ldbase.1723223135.93da.

### 3.2 OBJECT/FILE NAME

File names include:

- TP1_FINAL_public_V5.csv
- TP2_FINAL_public_V3.csv
- TP3_FINAL_public.csv
- TP4_FINAL_public_V3.csv
- haircortisoltp1and2.csv
- Master Codebook Teacher Job and Stress Study for sharing 2025.06.12_noendnote_1.docx
- Master Codebook Teacher Job and Stress Study for sharing 2025.06.12_0.pdf

### 3.3 DATA TYPE

The primary data are cleaned at the item level. All available composite and reverse scores, as indicated by standardized measures, are provided.

### 3.4 FORMAT NAMES AND VERSIONS

The dataset is accessible in the non-proprietary .csv format. The codebook is in .pdf and .docx format, which can be read by software such as Microsoft Word or Google Docs.

### 3.5 LANGUAGE

The data and documentation are stored in American English.

### 3.6 LICENSE

The license type for the dataset is ODC-By, which requires everyone who uses the dataset to cite the data.

### 3.7 LIMITS TO SHARING

The de-identified data are fully shared and are not under embargo.

### 3.8 PUBLICATION DATE

The data were first published on 07/02/2025.

### 3.9 FAIR DATA/CODEBOOK

The codebook and data are stored on LDbase, described with rich metadata, and are openly accessible. The codebook includes constructs measured, instruments and items assessed, variable names and labels (data dictionary), and scoring rules (e.g., reverse scoring). The data include most total scores for instruments and are stored in .csv format.

## (4) REUSE POTENTIAL

The teacher job and stress data are a valuable resource for others to reuse for additional research and teaching. The dataset includes many constructs of stress and stressors throughout an academic year during a historical moment in time: before, during, and after the surge of the omicron variant during the COVID-19 pandemic.

For research, investigators could use the dataset to continue investigating the influences of stressors on early educators' stress over time. For example, researchers could examine the effects of educators' job demands and resources on educators' stress, in line with JD-R models (Bakker & Demerouti, 2007; Kwon et al., 2021). Work could also explore concepts related to educators' allostatic load (McEwen & Wingfield, 2010), for example, by conceptualizing a theory-based composite of educators' psychological stress and examining its correspondence to physiological stress reactions. As a final example, researchers could probe the effect of timing on educators' stress, including its changes across the academic year and in relation to historical policy changes or stressors. For example, previous research has shown that the timing of stressors

Bates et al. *Journal of Open Psychology Data* DOI: 10.5334/jopd.134

and perceived occupational burnout may influence workers' physiological stress responses (Bärtl et al., 2023).

For teaching, educators could leverage the dataset to teach students about open science practices and various statistical analytic techniques, from descriptive statistics to analyses with repeated measures. For example, teachers could use the dataset in teaching demonstrations, as a basis for homework, replication examples, or other exploratory methodological work. Further, the dataset, accompanying codebook, and published paper can serve as an open-source and FAIR project example for students to follow when creating their own published data and codebooks.

When reusing this data, we would like to acknowledge five limitations of this data and the data collection process, as well as some strengths of the data. First, the study was limited to a small geographical area in the United States and used a nonprobability sample, which does not represent all early childhood educators. Thus, the data here, while informative, have limited generalizability. Second, self-reporting psychological health conditions may include some measurement error (e.g., social desirability bias). Yet, a strength of the study was supplementing these self-report measures with an objective estimate of physiological stress. Third, unmeasured variables may also account for our findings. For example, we did not measure all aspects of workplace stress, workplace stressors, and resources to fully demonstrate the theoretical JD-R conflict between educators' low resources and high demands. Fourth, missing data in our study, including from non-responders, may have also limited interpretation to those who completed the full measures. Notably, the sample size became more limited across the academic year. To address this limitation, researchers could use missing data strategies such as data pooling from other educator-focused datasets with similar measures (e.g., Kentucky Professional Development Framework Study [Rous & Grisham Brown, 2010], Cincinnati Preschool Promise study [Purtell et al., 2021]), bootstrapping, and multiple imputation. Fifth and finally, we conducted this study during a unique phase of the COVID-19 pandemic, around the start of the 2021–2022 academic year, which is a unique moment in history and likely not representative of the typical experiences of center-based early childhood educators. Despite our study's limitations and observational nature during an unprecedented time in history, this research still provides a nuanced understanding of educators' stress and stressors. Additional strengths of this data and data collection include the repeated measures of chronic physiological stress measures, estimated with hair cortisol concentration.

In conclusion, this dataset with a sample of center-based early childhood educators contains rich data on numerous constructs of their stressors and stress. With the publication of this dataset and accompanying paper, researchers and educators could have a valuable resource to advance science, theory, and education.

## FUNDING STATEMENT

In 2021, this work was supported in part by SproutFive, a Rabinowitz Award from the University of Cincinnati College of Nursing, the University of Cincinnati Office of Research, and the Center for Clinical and Translational Science and Training grant (2UL1TR001425-05A1). The content is solely the responsibility of the authors and does not necessarily represent the official views of the funders.

## COMPETING INTERESTS

The authors declare no financial conflict of interest associated with the publication of this manuscript. At the time of the study, co-investigator Dr. Jaclyn Dynia was involved in a research administrative position at one of the teacher recruitment sites, but had zero supervisory role of early childhood educational staff. As such, as a precaution, Dr. Dynia has never had access to identifiable information of the educators in the dataset.

## AUTHOR CONTRIBUTIONS

**Randi A. Bates** (Assistant Professor at the University of Cincinnati): Conceptualization, methodology, software, validation, formal analysis, investigation, resources, data curation, writing – original draft, writing – review and editing, visualization, project administration, data cleaning.

**Jaclyn M. Dynia** (Research Scientist at The Ohio State University): Conceptualization, methodology, investigation, resources, data curation, writing – review and editing, supervision, project administration.

**Bailey E. Martin** (PhD Candidate at the University of Cincinnati): data cleaning, writing – review and editing.

## AUTHOR AFFILIATIONS

**Randi A. Bates** orcid.org/0000-0002-6672-8155
University of Cincinnati, United States

**Jaclyn M. Dynia** orcid.org/0000-0002-4878-3791
The Ohio State University, United States

**Bailey E. Martin** orcid.org/0009-0009-1897-4894
University of Cincinnati, United States

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

## PEER REVIEW COMMENTS

*Journal of Open Psychology Data* has blind peer review, which is unblinded upon article acceptance. The editorial history of this article can be downloaded here:

- **PR File 1.** Peer Review History. DOI: https://doi.org/10.5334/jopd.134.pr1

**TO CITE THIS ARTICLE:**

**Submitted:** 26 February 2025    **Accepted:** 11 July 2025    **Published:** 23 July 2025

