## [Peer Review History. · Journal of Open Psychology Data]

Subject: [JOPD] Editor Decision - "Data from Early Childhood Educators' Work and Stress Study"

Dear Dr. Randi A. Bates, Dr. Jaclyn M. Dynia, Bailey E. Martin,

Thanks for your patience whilst we've undergone the review process. After review, we have reached a decision regarding your submission to Journal of Open Psychology Data, "Data from Early Childhood Educators' Work and Stress Study". The two reviewers have provided insightful comments that whilst supportive of the work, suggest some revisions may help with the clarity of communication to the audience. As such, our decision is to request revisions of the manuscript prior to acceptance for publication.

Instructions for how to resubmit your article online are pasted below. Please ensure that your revised files adhere to our author guidelines, and that the files are fully proofed prior to upload. Please also include a revised version of your article with 'tracked changes', adding comments where appropriate, to indicate the revisions made, in addition to a brief document outlining how you have responded to the reviewers' requests.

If you have trouble processing the revisions, our Help Center (<https://help.u-community.io>) or downloadable PDF (<https://bit.ly/Author-Guide-OJS-3>) may be able to help. If not, please get in touch and we'll be happy to help.

Please also ensure that all copyright permissions have been attained for any figures/tables you have included.

Please could you have the revisions submitted with two weeks. If you cannot make this deadline, please let us know as early as possible.

Kind regards,

Prof Thomas Rhys Evans

Reviewer D:
Recommendation: Revisions Required

Dear Drs. Bates and Dynia, and Ms. Martin,

It was my pleasure to have read your manuscript describing the publicly available dataset for Early Childhood Educators' Work and Stress. The dataset clearly has potential for reuse to advance knowledge about working conditions of the EC workforce. I have several comments aimed to improve the description of the dataset and one comment regarding the datasets themselves.

Regarding the paper itself

Abstract: If possible, include the main constructs available in the datasets in the abstract (e.g., self-efficacy, financial stressors, job satisfaction). I would recommend re-writing the last statement to refer to reuse potential that might provide insights (technically data doesn't provide insight, but analyses might?).

Introduction & Background: This was well written and gave a short overview of both the literature and theoretical frameworks.

Methods: The sampling, sample, and data collection section could use two points of clarification. First, how many individuals consented to participate (I thought 71, but it was unclear where the 4 withdrawals came from, and if they were different from the 3 listed in the table. Second, what is the attrition across the time points? Or differently stated, did some participants miss certain timepoints, or drop out all together?

I appreciate the detailed demographic section here, especially since some of these were deleted from the public data sets.

Quality control: I would have liked to see some more details regarding specific actions taken by each of the personnel involved. For example, did Qualtrics or RedCap design include quality checks (e.g., only integers allowed, only valid range number allowed). How were Qualtrics and RedCap files combined, and how were survey questions distributed across both platforms?

Data anonymization: Can you specify each variable that was eliminated? I understand no other de-identification methods were used, such as crosstabulation or truncation?

Data Type: Are all datasets considered primary data?

Reuse Potential: I think the reuse potential can be expanded and become more detailed. I'm thinking of looking back to the theoretical framework mentioned, and what type of questions other researchers could answer, or how this work might connect to other work on working conditions. I'm also wondering if there is potential for data aggregation, and if you have identified possible datasets with similar variables?

Regarding the datasets themselves, it seems the dataset for TP1 is incomplete. Only 28 participants were included, with a few missing study IDs. The dataset also only seemed to include Job Stress (js) and Self-efficacy (se) variables. Please make sure that the correct dataset is provided on LDbase.

Otherwise, the datasets are all in csv files, which is appropriate; and sufficient meta-data has been supplied to LDbase, which ensure findability. All datasets have a DOI attached, making citations easy to track under the ODC-BY license, I appreciated the codebook with its details. You may consider included a general statement on how missing data is indicated within the files (I assumed blanks, but it never hurts to be explicit). I would also suggest

supplying the codebook in pdf format, as it has more open-source options for reading.

Finally, I found a few typos:

WAQ p.8 – “not like my idea at all” (idea \diamond ideal?)

Reuse potential p.12 “the study was limited a small geographical area” (limited a \diamond limited to a?)

Funding statement p.13 “University of Cincinnati College of Nursing in the University of Cincinnati Office of Research” (Nursing in \diamond Nursing and?)

I hope these notes are helpful in revising this data paper, but please feel free to ask for clarifications if necessary.

Signed, Willa van Dijk

Reviewer G:

Recommendation: Revisions Required

The dataset has been appropriately reported and stored in accordance with the FAIR principles, demonstrating both accessibility and transparency in its documentation. Participant confidentiality has been safeguarded, with any data posing a potential risk to anonymity excluded from the shared dataset. This process is clearly detailed in the master codebook.

While the dataset is generally well-prepared and ethically managed, a few minor issues merit attention:

1. The names of some data files for specific time points (TPs) appear to be written differently in the manuscript compared to the actual file names stored in LDbase. For clarity and consistency, it would be helpful to ensure that file names referenced in the manuscript match those in the repository.
2. To enhance reuse potential, it would be helpful if the authors could clarify the specific types of secondary research the dataset could realistically support e.g., replication of certain trends, teaching demonstrations, or exploratory methodological work.
3. Although the value of longitudinal data is acknowledged, the reuse potential is considered limited due to the relatively small sample size, with participant numbers ranging from 41 to 67 across time points. The authors might consider discussing possible strategies to enhance statistical power or robustness (e.g., data pooling if possible, bootstrapping).
4. There are a few typographical errors throughout the manuscript. While I assume the manuscript will be professionally proofread before publication, I have marked some examples using tracked changes (e.g., on Page 9, in the paragraph under Chronic Physiological Stress Confounders, there is a typo and “Educators” should not be capitalised. There are also occasional extra tab spaces, which the authors can identify by enabling formatting marks. Although I have made some corrections, I recommend a thorough final check to ensure consistency and accuracy.
5. On Page 3, in the paragraph under Time of Data Collection, please consider adding the

year “2021” after “December” for TP2 (i.e., “December 2021”) to avoid any potential confusion about the timing.

6. While the dataset is openly available, the README file (Master Codebook) does not currently specify the licensing terms. The authors should consider including a clear reference to the ODC-By license within the Master Codebook to enhance transparency and support appropriate reuse.

7. There is a notable decrease in survey participation from 67 participants at TP1 to 53 at TP2, representing a drop of 14 participants. Similar dropouts are observed at other time points for both survey and hair data. The manuscript only mentions four withdrawals, leaving the reasons for the attrition unclear. Providing a detailed explanation of participant dropout or non-response between these time points would improve the transparency and interpretability of the dataset.

8. Although gender data were not reported due to confidentiality, the demographics table in the manuscript includes the percentage of female participants. The authors might consider cross-referencing this information with the Master Codebook (perhaps as a note) to ensure consistency and clarity.

9. The authors should explicitly state which variables were removed due to confidentiality or other reasons, or indicate in the manuscript how this information can be verified in the Master Codebook (e.g., notes added in the tables or cells corresponding to those variables). Note that page numbers have been added to the manuscript to facilitate ease of reference during the revision process.